# Tuberculosis and Migrant Pathways in an Urban Setting: A Mixed-Method Case Study on a Treatment Centre in the Lisbon Metropolitan Area, Portugal

**DOI:** 10.3390/ijerph19073834

**Published:** 2022-03-23

**Authors:** Rafaela M. Ribeiro, Luzia Gonçalves, Philip J. Havik, Isabel Craveiro

**Affiliations:** 1Global Health and Tropical Medicine (GHTM), Instituto de Higiene e Medicina Tropical (IHMT), Universidade Nova de Lisboa (UNL), Rua da Junqueira 100, 1349-008 Lisboa, Portugal; luziag@ihmt.unl.pt (L.G.); isabelc@ihmt.unl.pt (I.C.); 2Centro de Estatística e Aplicações da Universidade de Lisboa (CEAUL), 1749-016 Lisboa, Portugal

**Keywords:** tuberculosis, migrants, case study, urban health, health inequalities, social determinants, Portugal

## Abstract

Tuberculosis (TB) is an infectious disease associated with poverty. In the European Union TB tends to concentrate in urban settings. In Lisbon, previous studies revealed, the presence of migrant populations from a high endemic country, is one of the risk factors contributing to TB. To better understand TB in foreign-born individuals in the Lisbon Metropolitan Area, a mixed-method case study was undertaken on a TB treatment centre in a high-risk part of urban Portugal. Quantitatively, annual TB cases were analysed from 2008 to 2018, dividing foreign-origin cases into recent migrants and long-term migrants. Qualitatively, we explored recent migrants’ reasons, experiences and perceptions associated with the disease. Our results showed that foreign-born individuals accounted for 45.7% of cases, mainly originated from Angola, Guinea-Bissau, and Cabo Verde. TB in recent migrants increased over the years for Angola and Guinea-Bissau, while for Cabo Verde TB cases were due to migrants residing in Portugal for more than 2 years. Recent migrants’ reasons to travel to Portugal were to study, to live and work, tourism, and seeking better healthcare. Visiting family and friends, historical links and common language were key drivers for the choice of country. Recent migrants and long-term migrants may present distinct background profiles associated with diagnosed TB.

## 1. Introduction

Although it is treatable and can be prevented, Tuberculosis (TB) remains one of the deadliest communicable diseases globally, causing around 1.4 million deaths in 2019 [1]. It is an air-borne infectious disease with a slow progression [2], typically affecting the lung; nevertheless, 16% out of 7.1 million notified cases worldwide in 2019, corresponded to extra-pulmonary TB (EPTB) [1]. TB’s daily treatment lasts, at least 6 months. However, treatment for EPTB in bones or meninges, may be prolonged from 9 to 12 months, and for antibiotic-resistant bacteria, up to 2 years [3].

Infection with *Mycobacterium tuberculosis* does not mean falling ill with TB. It is estimated that approximately, 2 billion infected individuals worldwide remain symptomless from the disease (latent TB), and only 5 to 10% will develop the disease during their lifetime [1]. The human immunodeficiency virus (HIV) strongly affect the expression of disease in infected individuals [4].

Risk factors, at the individual level, are linked to impaired host defence, such as HIV, malignancies, immunosuppressive treatment, malnutrition, smoking, diabetes, alcohol abuse and indoor pollution [5]. At the population level, living and working conditions, such as overcrowding and poorly ventilated environments, strongly affect exposure to TB, increasing the risk of TB transmission [5]. It is broadly recognised that impoverished sections of society are more heavily affected by TB burden, as they are more exposed to individual and population risk factors, and may face increased difficulties in following treatment [6].

As a recognised worldwide public health problem, TB is the focus of the World Health Organisation’s (WHO) End TB strategy, setting ambitious goals for incidence and mortality reduction by 2035 [7]. Expectations for disease control are viewed as overoptimistic by some [8]. M. Gomes et al. debate methods ignoring heterogeneity of individual characteristics, such as exposure to a causative agent or susceptibility given exposure, may lead to underestimating overall population risk from disease. In this sense, inequalities should be reduced, for all individuals in a population face similar risk of disease fostering population-wide interventions for TB control [9].

Inequalities defined as “the state of not being equal, especially in status, rights and opportunities” touches upon a broad range of issues from material wealth distribution to unequal access to opportunities based on sex, age, disability, origin, religion, or ethnicity, amongst others [10,11]. Inequalities directly affect people’s health and have been object of attention by the United Nations with the Sustainable Development Goal (SDG) number 10 [11]. SDG 1, 2 and 3 also relates to TB. These goals enhance the urge to tackle underlying socio-economic causes of disease burden in a common global perspective. The impact of social protection and poverty elimination on TB control, resulted in a reduction by 84.3% of disease incidence in a statistical modelling analysis [12]. 

TB distribution is hugely unequal around the world. Thirty countries concentrate around 90% of the TB disease burden-none of them a high-income country [1]. The WHO Region of Europe hosted 2.5% of TB cases in 2019 with a total best incidence of 26 (uncertainty interval of 23–30) cases/100,000 inhabitants [1]. For European Union (EU) countries, the notification rate was of 9.6/100,000 inhabitants in 2019, and approximately a third of TB burden in the WHO European region was attributed to migrants [13]. Nevertheless, disparities among countries on how to account TB in foreign-born individuals (e.g., birthplace or citizenship) may generate difficulties in monitoring migrant’s health [14]. Portugal records foreign TB cases by birthplace [15]; foreign TB contribution has been increasing, although remaining lower than the EU, from 18.3% (2016) to 23.7%(2019) [13,15]. 

Resident foreign-born individuals in Portugal have increased in 34.1% between 2015 and 2019, resulting in a total of 590.348 persons in 2019 [16] accounting for 5.7% of the resident population in the country (using 2021 census denominator) [17]. In the context of population mobility, infectious diseases have the potential to affect host countries´ epidemiology [18]. Individuals migrating from high TB burden countries, might suffer from infectious TB at arrival [19], or carry a latent infection with a potential to convert into disease, especially in circumstances of difficult living conditions [20]. In a Canadian study, the initial period following arrival, is where the expression of TB may predominantly occur; and immigrants from Sub-Saharan Africa had the highest rates for both sexes [21]. 

As a result, the place of origin of a migrating person is relevant, particularly for recently arrived individuals. Language barriers and difficulty to access healthcare may promote diagnostic delays and, in the case of communicable diseases, prolong infectiousness in the community [22]. A spatial analysis at the individual level, showed significantly longer TB diagnostic delays among foreign-born individuals from high TB endemic countries, for the Lisbon area [23]. Another study based upon interviews with Portuguese TB healthcare providers concludes that limited socioeconomic resources, complex bureaucracy at the entry point, and obstacles for social protection affected TB illness presentation in foreign-born patients [24]. 

Geographical maldistribution of wealth generates inequalities between countries, inside a city, a municipality or neighbourhood, influencing disease epidemiology at a local level. Impoverished areas, typically inserted in urban settings, present high population density with poor housing conditions, and are usually affected by inadequate public health responses, which favours TB emergence and spread [25]. These areas may host individuals accumulating risk factors, associated with socio-economic hardship, namely drugs-users, alcoholics, immigrants from high burden countries, homeless, and prison inmates [25,26]. This spatial clustering of TB risk factors may create “hyperendemic areas” generally called disease “hotspots” [27]. In the EU, where TB incidences are relatively low, the TB burden tends to be concentrated in big cities, presenting higher incidence rates than the national average [25,28]. 

Big cities may host a higher number of foreign-born individuals [29]. In the case of Lisbon, 7 out of 10 municipalities registering the highest number of foreign-born resident individuals, were located in the Lisbon Metropolitan Area (LMA) [16]. Amadora and Sintra are two municipalities of the LMA registering a high number of foreign-born resident citizens [16]. The distribution of nationalities varies at national level, and at a municipality level (Figure 1). 

The municipality of Amadora and Sintra, both examples of multi-cultural urban settings, present a singular profile of foreign-born resident citizens (Figure 1). Other municipalities of the LMA may present different nationality profiles. Amadora hosts impoverished urban areas [30], and is considered a municipality presenting an accumulation of TB risk factors [26]; Sintra as an adjacent municipality, present similar neighbourhood profile [31]. Geographical distribution of TB risk factors varies among municipalities for pulmonary TB [26]. One of TB risk factors for this area is being a migrant from a high TB burden country [26]. In this sense, not only the local living conditions may influence TB expression, but also, the epidemiological TB profile of the countries of origin of foreign-born residents (Table 1).

In Portugal, TB is treated in specific centres–Centro de Diagnóstico Pneumológico (CDP)—which are ambulatory public healthcare institutions responsible for screening and treating individuals with TB (latent TB and disease). Each CDP serves a pre-stablished geographical area. TB treatment is given free of charge. Once a case is confirmed, patient´s data are inserted in the National Surveillance System for Tuberculosis (SVIG-TB). 

The Portuguese national health service—Serviço Nacional de Saúde (SNS)—provides a tendentially universal access mainly funded via taxation for its citizens and legal foreign residents, who access healthcare through a user number. In principle, every citizen is entitled to a family doctor in primary healthcare (PHC) who act as gatekeepers for specialised care. However, coverage is not yet complete, nor is homogeneous for the whole country, affecting accessibility to health services [32].

For irregular foreign-born citizens, when vital and urgent needs are at stake, or a public health concern, namely for infectious communicable diseases, vaccinations or maternal health, care should be provided regardless of legal status in equal circumstances compared to national citizens [33]. Additionally, some countries maintain bilateral cooperation agreements with Portugal, namely, the African Portuguese Speaking countries–Países Africanos de Língua Portuguesa (PALOP), which are invoked when human resources or technical competency in health, is not available in the country of origin, guaranteeing necessary healthcare in the host country [33]. 

This study is focused on one CDP serving an urban area of the LMA considered to have a severe high risk for TB [26]. Through a mixed-method case study [34], it aims to characterise foreign-born TB contribution. Firstly, we measured the contribution of foreign-born individuals to the burden of tuberculosis in eleven years (2008–2018). Secondly, focusing on the share of TB patients being diagnosed with TB within the first two years following arrival, we explored their reasons for coming to Portugal, and their experiences and perceptions associated with the disease. We aim to analyse, explore, and theorise about different TB disease pathways in foreign-born individuals, for a given context.

## 2. Materials and Methods

### 2.1. Study Setting

A mixed-method case study was conducted on one CDP localised in the north LMA (study site). This CDP served a pre-established geographical area comprising two municipalities—Sintra and Amadora—plus a portion of a neighbouring one—Odivelas, serving around 600.000 inhabitants [35] (Figure 2). Field work to collect official data and perform interviews was carried out between October 2018 and January 2019. At the time of the study, MDR-TB was not treated at the study site, nor other co-morbidities. The working staff during filed work included, one medical doctor, one nurse, two secretaries, and one radiology technician. 

The municipality of Amadora is a densely populated urban area, hosting a total of 171,500 inhabitants (census 2021) [36], and 17,797 foreign-born residents in 2018 [37] (~10.4%); its population density being 7641.9 inhabitants/km^2^ for the same year [17]. Sintra hosted 385,654 inhabitants 8.2% of which foreign-born registered residents. Sintra’s population density was 1216.8 inhabitants/km^2^ with 34.9% of its territory accounting as a protected natural area [17]. Comparatively, the municipality of Lisbon hosted 545,923 inhabitans [36], with a population density of 5069.7 inhabitants/km^2^ [17], 14.6% of which foreign-born residents [37].

In 2018, access to PHC with an assigned medical doctor, at a national level, covered 93.0% of citizens; for Amadora and Sintra, 26.1% and 20.5% of registered citizens lacked one, respectively. In Lisbon-city this proportion was of 11.2% [38]. Moreover, the municipality of Amadora hosted a “critical urban area*”*, defined as “territories considered critical from the point of view of spatial segregation and concentration of the phenomena of urban poverty, exclusion, and criminality as well as inconsistencies in terms of urban planning, among other elements that potentially stigmatize these neighbourhoods”. It was subject to governmental attention through incentives to its reform and development [30]. 

### 2.2. Study Design 

A case study is the ideal research method to study a phenomenon embedded in its “real life context” [34]. This case study is focused on a CDP carefully selected as a critical case–type 1: single case, with a single unit of analysis [34]. The choice of the studied CDP as a critical case, was based on one study [26], where spatial distribution of risk factors for pulmonary TB in Portugal was analysed. The chosen CDP serves the urban municipality where the accumulation of TB risk factors was the highest in Portugal. Hence, this CDP was considered a critical case. 

The Inverse Care Law by Hart T. (1970) [39] stands that the availability of good medical care tends to vary inversely with the need for it in the population served [40]. Considering TB and its risk factors associated with poverty [6], our working hypothesis is that, exploring TB pathways may shed light on healthcare seeking behaviour and healthcare provision to the more disadvantaged strata of society. Our premise is that more disadvantaged people have increased problems in receiving appropriate healthcare. The studied CDP situated in a TB “hotspot”, represents the phenomenon under consideration in its extreme form which is ideally suited to confirm an initial theoretical proposition or find alternative set of more relevant explanations [34]. In addition it offers a greater potential for generalisation [41,42]. 

Amongst other possible risk factors present in the study area, such as high HIV/AIDS incidence rate, overcrowding, non-standard accommodation, unemployment, and prison population [26], this paper focused on being a foreign-born citizen from a high TB burden country. In informal conversations with the gatekeeper–the medical doctor responsible for the CDP, he expressed a perception in his practice supported by internal reports, of an increased number of recently arriving foreign-born TB patients, at the study site. We considered this information a relevant expert´s opinion and following an exploratory operational research mindset, we chose to study this topic in depth. 

### 2.3. Data Collection and Analysis

We performed a single case study with a mixed method approach, using a convergent design [43]. Qualitative and quantitative data were collected simultaneously, analysed independently and triangulated to build in depth knowledge around the case [44]. 

The quantitative data relies exclusively on the SVIG-TB for the specific CDP subject of study. The number of notified TB cases from 2008 to 2018 was analysed by descriptive and exploratory statistics. In the case of foreign-born patients, the SVIG-TB included, among other variables, the country of origin, the year of arrival, and if TB diagnosis was made within two years following arrival (yes or no). 

We obtained lists of the total number of foreign-TB cases, per year, and lists of total number of cases diagnosed within the first two years following arrival, per year. Both lists discriminated origin countries of individuals. Information was analysed. Based on SVIG-TB variables, we empirically adopted the following definitions to describe the results:Migrant—a person who is born outside Portugal independently of legal status. In Portugal TB in foreign-born citizens is accounted by country of birth.Recent migrant—a migrant being diagnosed with TB within a 2 year-period following arrival in Portugal [TB diagnosis within two years following arrival (yes)]Long-term migrant—a migrant being diagnosed of TB after 2 year-period following arrival in Portugal. [Total TB foreign-born per year minus TB diagnosis within two years following arrival]Autochthonous TB–TB in Portuguese-born individuals.

The qualitative data was collected through semi-structured interviews. Recent migrant patients constituted a purposive sample from the consultation list of the field work months. Inclusion criteria were being diagnosed of TB within the 2 years following arrival, 18 years old or over, and undergoing active treatment for any type of TB. No exclusion criteria were previously established. The gatekeeper´s opinion was considered when inviting patients.

The interview guide was pilot tested with the first interview when field work started, and minimal changes were made. Themes explored centred upon usual healthcare circuits in the countries of origin, initial symptoms of TB and health seeking behaviours, healthcare circuits in Portugal, and the experience of being diagnosed with TB. Topic guide for the semi-structured interviews can be found in Appendix A. 

All interviews were done by the first author (RR), took place after TB consultations, in a private room inside the CDP, face-to-face. We interviewed 13 participants out of 20 purposely selected, including the pilot interview. Reasons given for declining participation were refusal to give consent after talking briefly about the interview topics, lack of time or interest in participating, a bad health status, and not showing up for the consultation. When permitted interviews were audio-recorded and, otherwise written notes were taken (two interviews). Interviews lasted around 40 min. 

Interviews were transcribed verbatim by RR. Initial thematic qualitative data analysis [44] was done simultaneously and independently by RR and IC who agreed on basic codes and themes. Further coding was done by RR. Data was analysed deductively following a constructivist approach where perspectives of experiences, and perceptions of people, were valued in search for a deeper understanding of the phenomenon. Themes identified were (a) reasons to travel, (b) accessibility to healthcare in Portugal, (c) the choice of country, (d) healthcare in home countries, (e) the experience of TB diagnosis and (f) enabling factors. We used the consolidated criteria for reporting qualitative research when possible (COREQ) [45].

### 2.4. Ethical Considerations

Ethical approval was granted by the Lisbon and Tagus Valley Regional Health Administration of-ARS-LVT-(ref.11703/CES/2018). Participants voluntarily signed a written informed consent to participate and did not receive any financial compensation. All files are stored safely until the project is concluded whereafter raw material will be destroyed following ethical requirements of the approved research protocol. Personal information was anonymised when first analysing the information. 

## 3. Results

### 3.1. Secondary Quantitative Data Analysis

Notified cases from SVIG-TB, for the studied area, were analysed, by countries of birth, for an eleven-year period (2008–2018) (Table 2). Countries of origin accounting for a total of more than 20 notified cases during the studied period were tabulated. Other countries showing 20 notified cases or less were grouped as “Others”; India and Romania presented 10 to 20 cases; Guinea-Conakry, Ukraine, and Pakistan presented 5 to 10 cases; and Canada, France, Timor, Philippines, Bangladesh, China, Congo, Moldavia, UK, Italy, Senegal, Russia, Nepal, Germany, Gabon, Leetonia, South Africa, Andorra, Spain, Gambia, and Venezuela, presented 1 to 5 notified cases. 

Portugal, presented a total of 1031 cases for the studied period, accounting for 54.3% of all notified cases (Figure 3). The countries contributing the most to cases of foreign origin were Guinea-Bissau, Cabo Verde, and Angola presenting totals of 253, 226 and 214 cases respectively which accounted for 36.5% of all cases. Each country contributed with a similar share of cases (Figure 3). As a group of foreign countries contributing the most to the TB burden in the studied area, they presented a mean of 63 cases/year between 2008 and 2018. The lowest number of cases was of 11 for Cabo-Verde in 2018, and the highest 31 cases for Angola, in 2017. In 2018, cases decreased for Portugal and the other 3 countries. Cabo Verde and Portugal registered the lowest number of cases in 2018. 

São Tomé and Príncipe, Brazil, and Mozambique presented a lower number of notified cases, totalling around 30 cases each for the entire period under consideration. Each of these countries presented between 0 to 7 cases, per year; and as group presented a mean of 8.7 cases/year (2008–2018). All countries included in Table 2 speak Portuguese as their official language. 

Angola, Guinea-Bissau, and Cabo Verde presented the greatest contribution for notified TB cases in foreign-born patients in the studied area (Figure 3). Time-line graphs shown in Figure 4 quantify the evolution of the proportion of TB cases in recent migrants, out of the total number of cases, for the aforesaid countries.

Guinea-Bissau and Angola presented a gradually increasing number of notified TB cases in recent migrant patients for the studied period. For Guinea-Bissau 31.6% of its TB cases occurred in recent migrants for 2009; this proportion decreased sharply afterwards (4.5% in 2010) followed by a slow sustained increase in its number, until 2016 when it reached 52.0% of cases. We may infer that TB patients from Guinea-Bissau recently arriving in Portugal were potentially modifying the epidemiological TB profile at a local level, as a new influx of cases. However, the absolute number of cases did not show a marked rise (Table 2). Although the proportion of TB cases in recent migrants were rising, this suggests it did not manifestly affect the absolute number of cases, probably because TB cases in long-term migrants were decreasing. 

Angola, similar to Guinea-Bissau, presented an increasing number of recent migrant TB cases from 2009 onwards. For Angola, the proportion of notified TB cases in recent and long-term migrants tended to converge in 2016, in a 50–50 proportion, similar to Guinea-Bissau. In 2018, 65.0% of notified TB cases, in Angolan-born individuals, were in recent migrants. The absolute number of TB cases, for Angola rose in 2016 and 2017, when compared to previous years (Table 2). For the same years we also observe an increase in the proportion of recent migrant TB cases being notified. This fact suggests the total rise in TB cases for this country, are probably due to cases in recently arrived migrants (Table 2). 

Cabo Verde illustrates a completely different pattern given that notified cases were predominantly attributed to long-term migrants for the entire period, the maximum proportion of recent migrants being 21.1% in 2015. 

In 2018, Cabo Verde reached its lowest number of absolute cases coinciding with the lowest number of notified autochthonous TB. Individuals from Cabo Verde were long-term migrants, and despite a rising tendency of notified TB cases in recent migrants for Angola and Guinea-Bissau (Figure 4), the absolute number of cases was lower, when compared to the previous two years (Table 2). This fact suggests that in the studied site and in relation to the contribution of these 3 countries despite a rising case load in recent migrants from Angola and Guinea-Bissau, the overall TB notification rate declined, in 2018.

### 3.2. Primary Qualitative Data Analysis 

Aiming to understand *recent migrant* TB patients´ motivation to come to Portugal, semi-structured interviews were held during field work. All participants listed for invitation (20 persons) were from Angola or Guinea-Bissau, except one from Guinea-Conakry. We interviewed 13 recent migrant TB patients in active treatment. 

Most participants presented extra-pulmonary TB (9/13), were female (8/13), and from Angola (9/13). Participants were aged between 19 and 36. Four participants had HIV (Table 3). In Appendix B detailed information is provided on interviewees’ transnational TB healthcare pathways. HIV positive TB patients were following active treatment for HIV in another healthcare service while in Portugal.

**Reasons to travel.** Four main reasons given by participants to come to Portugal: (a) to study, (b) to live and work, (c) to travel as a tourist, and (d) in search of better healthcare than in their home countries. Most participants came because of health-seeking behaviour (8/13) (Table 3). Participants coming for reasons other than healthcare were a master student who fell ill 6 months after arrival, a tourist coming to buy clothes/gadgets who suddenly fell ill, another tourist having a pre-booked flight who sought medical care upon feeling ill, a teenager who came to live with his family, another registered as a refugee, and finally one participant who had lived in Portugal previously and decided to return. 

Health was a key reason for participants to travel (Table 3). Young participants, arriving in Portugal in pursuit of better healthcare, explained a pathway marked by diagnostic uncertainty and a progressive state of illness, motivating them to travel.

*“They could give me medication not appropriate for me… I have heard of people… taking medication not appropriate for their pathology and having consequences, sometimes even more serious… it was the case of my father, he almost died… these issues I have seen there, made me risk everything and come to check what is going on with me, to be sure. Then I thought about Portugal”*.(P1)

Two participants opted for experimenting healthcare in neighbouring countries, namely Senegal or Namibia, before travelling to Portugal.

*“(…) In Senegal… they gave me treatment and I felt better, I thought everything was fine, but then I continued with pain, it got worse, I could not even get out of bed in the morning, walk, or work, and then I came here”*.(P4)

**Accessibility to healthcare in Portugal.** Six participants had an SNS user number (6/13) (Table 3). Two participants had previously lived in Portugal for many years and returned. Migrant participants arriving on a tourist visa who become irregular after it expires, without a user number, were being treated for TB, as it is considered a vital/urgent healthcare need. 

Entry points in the SNS were diverse, and preferences were motivated by out-of-pocket fees (in PHC out-of-pocket fees were perceived to be lower than hospital setting), informal knowledge (master student familiar to the hospital setting associated with his studies), financial capacity (private service) or availability of service (ambulance). No participant accessed the CDP service directly without a previous referral. 


*“If it wasn’t through Primary Health Care it would be too expensive, and I could not make it…”*

*[P10 took advice from a friend]*


Additionally, some participants preferred to use private healthcare in Portugal. Two participants even showed familiarity with the Portuguese private healthcare service accessing medical consultations whenever coming as tourists.


*“Private clinics… there you go ask for an appointment, the doctor comes register things, you go, you have results, the doctor explains it, then you go on with other things…”*
(P8)

**The choice of country.** The reasons presented for choosing Portugal were visiting family and friends (VFF), common language, historical links, and bilateral agreements between countries. One participant from Guinea-Bissau followed all the bureaucratic procedures required for healthcare bilateral agreements with Portugal.


*“I chose Portugal because here I know how to speak the language, and also because it is the country which colonised Guinea-Bissau, I have more rights here than in other countries…”*
(P3)

Speaking a common language was stressed by participants; one HIV positive participant with family in France preferred to come to Portugal because of the language barrier. She suffered disease related stigma and was unable (or unwilling) to disclose diagnosis to family members, acting as translators. 

Some participants first visited Portugal because of this illness episode, and most of them were VFF. Some participants showed a manifest preference for Portugal, based on stories of cured returnees. 


*“I thought about Portugal as the easiest… I have family here and the local language is Portuguese… there is a connection… we have a very good impression as people doing consultations return, cured, healed… I saw myself in this condition, I came to have a good consultation and get cured”*
(P1)

Portugal also emerges as a final option, instead of facing possible fatal outcomes in their home countries.


*“I was not so worried, as if I were in Angola… I would have thought “I will die tomorrow”. As I was here, I did not worry… I felt taken care of…”*
(P2)

*“There… if you don’t have someone in the family who is a doctor, or someone important… you die easily”*.(P7)

**Healthcare in home countries.** Some participants explained low usage of medical services in their home countries, expressing a lack of trust in their healthcare systems. The public healthcare systems of Angola and Guinea-Bissau were perceived as being of low quality, affected by a lack of essential equipment such as gloves, syringes, needles, or equipment to perform diagnostic exams. Some participants explained the need to pay for any material requested in the hospital. 


*“Many times, we just have the consultation itself… there is no medicine in the hospital, and sometimes even syringes, needles, alcohol to disinfect… patient’s family has to buy it in the pharmacy out of the hospital and it is very complicated… we must buy all disposable materials”*
(P5)

Health professionals were seen to be exhausted and not motivated, leading to patient mistreatment. One participant complained about a lack of follow-up in clinical history with frequent repetition of exams and treatments. 

According to some interviewees, private healthcare in their home countries, was of better quality, because of more caring professionals, better quality of exams and infrastructures, and more time per consultations. 

*“To give birth I prefer the clinic, there is a better service because you are going to pay… in the public hospital there is no service… they respond badly to people, they don’t help people”*.(P4)

Nevertheless, for this illness episode, even for participants able to afford private consultations, diagnostic uncertainty remained, illness advanced, pushing them towards other alternatives, such as travelling to Portugal. Moreover, even when TB was suspected or diagnosed, participants explained difficulties with local TB pathways. 

Four participants from Angola initiated TB treatment but faced major difficulties in acquiring the drugs, owing to high prices and shortage of medicines, leading to treatment interruption. In contrast, antiretroviral treatment for HVI was not difficult to obtain, according to two Angolan HIV-positive participants.

*“In Angola it was difficult… to acquire medicines was a fight… to get it I would visit several pharmacies, contact anonymous resellers… the prescribed medicines were not available in the pharmacies. Sometimes I would remain 2–3 weeks without medication… and when we had the drugs, medicines were very expensive”*.(P1)

*“In this hospital… medicine must be given to patients for free, but I think they sold it… you have to do “business” with nurses or technicians … they themselves are the ones taking from the hospital and selling”*.(P7)


*“To get medicines is the most difficult, people say come “tomorrow”, “tomorrow”, and if you pay, there is medication, especially in the case of TB, because HIV drugs are still for free”*
(P10)

**The experience of TB diagnosis.** In Portugal, TB diagnosis was a major surprise for most of them. In participants’ imagination, TB was associated with an unruly lifestyle, bad eating habits, and cough. Most participants struggled with diagnostic incertitude, not only in their home countries, but also along the diagnostic care cascade in Portugal; particularly for EPTB. Some participants had never heard about EPTB. 


*“I was not confident; I was not sure… I have seen people sick of TB… but the way it happened to me, to fall and faint, I have never seen it. I was not believing it… because TB on the head, I have never heard of it”*
(P8)

Perceived disease related stigma played a major role in participants’ narratives, leading some to hide diagnosis, even from family members. One participant explained warning her workplace about the need for contact tracing, without being taken seriously.


*“Until now they don’t believe it, first they start with my lifestyle-“She eats fruits, eats on time, she takes care” … then I did not have contact with infectiology... We are Africans… they prefer to look at other little things than believe X had TB… even with the report… it worries me until now. Myself I did not believe it was TB”*

*[P11, talking about her workplace]*


Another participant was confronted with TB diagnosis in her home country but did not believe it and pursued second opinions and further exams abroad. In Portugal, the same participant had an inaugural HIV diagnosis while hospitalised, however she avoided the issue during the interview. For HIV positive participants, their HIV status seemed emotionally more painful and associated with a worse disease-related stigma than TB diagnosis. 


*“I was very scared, lots of fear, of not holding on until here, because I went to the healthcare centre and when I arrived there, I did not say anything to the doctor [HIV diagnosis]. I just told it here. I just said I was sick of tuberculosis… I feel a lot of shame”*
(P3)

Many participants stated they were economically and/or emotionally unprepared to remain ill abroad for such a long period for treatment; some interviewees faced financial hardship during their stay. Despite causing them suffering, a sensation of time-loss and disorganisation of their lives, they prioritized their health. 


*“Everything was upside down, my daughter had to drop school and come here rapidly, financially we had to toughen up to be able to stay here”*
(P11)

**Enabling factors.** All participants were grateful for the healthcare received and mainly for the successful search for diagnostic certainty through complementary means of diagnosis, while in Portugal. They were also thankful for health professionals handling patients with care. Family support, good food and religious beliefs, allied with the provision of healthcare were the main motivations for participants to overcome the challenges of curative process. 


*“I cannot deny anything because they said everything was proven by laboratory exams, then I accepted it”*
(P4)


*“In Portugal it is better, there is variety, it is cheaper also [food]… For instance, here people eat meat every day, there is fruit with vitamins. In Guinea [Bissau] we eat meat every 2, 3 or 6 months… only fish, but of low quality. There are vegetables but no fruit. There, it is more expensive, even after money conversion”*
(P4)

## 4. Discussion

This article presents a mixed-method case study focused on a TB treatment centre serving a critical urban area; it includes a descriptive and exploratory analysis of notified cases between 2008–2018 and semi-structured interviews. The aforesaid study was based on the premise that TB healthcare pathways may illustrate the difficulties most vulnerable people face, such as migrants, in obtaining appropriate care.

The main findings of this study are: (a) autochthonous TB is the main contributor to notified cases of TB, (b) key countries contributing to foreign TB were Cabo Verde, Angola, and Guinea-Bissau, in similar proportions, however with different disease profile, (c) the proportion of cases in recently arrived migrants increased gradually over the years for TB patients from Angola and Guinea-Bissau, (d) interviewed participants considered healthcare systems of countries of origin weak and unresponsive to their medical needs, (e) seeking healthcare was a key motivation for travel according to recent migrant TB study participants; visiting family and friends, common language and historical links between countries strongly influenced migration options. 

Portugal is an endemic country for tuberculosis [46]. The urban studied area accumulates other TB risk factors affecting TB expression in autochthonous population [28]. Disease profiles differ between nationals and foreign-born individuals [47]. Ploubidis et al. argues ~50% of TB variation in the WHO European Region relates to a nation’s wealth and degree of egalitarianism-the richer and more equal a country is, the less TB cases [48]. Regarding medical care, Portugal expressed an unequal distribution of access to specialist medical care, when compared to 21 countries of the Organisation for Economic Co-operation and Development (OECD), with a pro-rich tendency [49]. This fact may affect medical services provision for the studied area, contributing to TB expression.

The highest number of notified TB cases were found in autochthonous patients, however as Portuguese-born residents out-number those who are foreign-born, TB indicators, such as incidence and prevalence, are most likely lower for them. Moreover, diagnostic delays and undetected cases also influence TB detection rates. In Portugal, research shows migrant population use healthcare services less [50], face more accessibility barriers [24], and access TB services in a more severe state of illness [24], suggesting the existence of detection gaps.

This case study provides evidence for hypothesising about two key origins of TB, “endogenous*”* and “exogenous*”*. Endogenous TB cases are supposedly associated with autochthonous TB and lower social economic status [51]. The poor face higher risk of acquiring the infection and developing the disease [6]. In our study, long-term migrants, with Cabo Verde serving as an example, most probably correspond to endogenous TB cases. 

Three reasons support this hypothesis. First, a study analysing the genetic profile of *M. tuberculosis* in the LMA argues in favour of endemic Portuguese strains likely to penetrate in migrant populations, namely from PALOP [52]. Second, at a country level, Cabo Verde, boasts good TB indicators and a treatment coverage of 95% (Table 1), thereby achieving the End TB strategy goal [1]; even if Cabo-Verdeans travelled to their home country, they would be unlikely to contract *M. Tuberculosis*. Third, it is possible that foreign-born population face increased difficult living conditions; an Italian study comparing socio-economic status of Italians and migrant TB patients (median of 7.6 years of arrival in Italy), concludes that 46% of migrants with TB were severely deprived, when compared with 9% of Italians [53]. Moreover, it is possible that the studied area experienced some degree of residential segregation of ethnic minorities [31] which boosts inequalities and potentially contributes to higher TB rates within the segregated area [54]. All arguments favour the hypothesis that infection and disease were associated with the host country.

On the other hand, TB may also be categorised as exogenous, through importing *M. tuberculosis* from high burden countries. If imported and exogenous, the disease is likely to develop during the initial years following arrival [21]. Imported cases, may less directly relate to individual social determinants of health. In a study done in the United States, low socio-economic status was weakly associated with TB in foreign-born individuals [51]. Our qualitative results show recently arrived migrant TB participants from Angola and Guinea-Bissau exhibiting purchasing power to travel and seek healthcare treatment abroad. 

Exogenous TB may strongly reflect the weakness of healthcare systems of origin countries. In the case of Angola and Guinea-Bissau, not only TB and TB-HIV incidences are very high, but also other specific TB indicators, such as treatment coverage, are very low (Table 1). Treatment coverage may serve as an equity-sensitive tracer of a country’s degree of social protection coverage and general healthcare access [55]. 

Health systems’ structural problems certainly affects TB management [56,57]. Evidence shows that Angola presents a probably higher multidrug resistance than estimated by WHO [58]; as does Guinea-Bissau [59]. TB burden in children following active contact-tracing of adult TB patients for both countries is worrisome [60]; in a cohort of 124 children contact of TB adult patients in Luanda, 56.5% had active TB and 20% were HIV positive [61]. Substandard TB drugs may exist in the private market in Angola [62], which is a known problem for low-income settings [63]. Participants in our study expressed their difficulties in accessing TB drugs. In a qualitative study with Angolan health professionals the existence of active black-market for TB drugs was suggested [57]. In Guinea-Bissau, a study shows how smear-negative TB patients (associated with HIV-TB coinfection), remain symptomatic and undiagnosed, presenting a high mortality [64]. 

Recent migrant TB study participants were young people from Angola and Guinea-Bissau with motivation and resources to travel to a foreign country. Quoting from a qualitative study with Angolan healthcare TB professionals: ‘Wealthy Angolan people who get TB will go abroad for treatment’ [57]. Moreover, some interviewees showed familiarity with Portugal, travelling for a few times before this illness episode. Pathways to healthcare seemed roughly clear for some, and participants navigating different circuits, used the private healthcare and/or the public healthcare system in Portugal. For some healthcare seeking behaviour was timely and access to the healthcare system in Portugal smooth, for others the journey towards TB diagnosis and treatment was longer and more painful, especially for EPTB [65].

Transnational routes for recently arrived migrant participants were facilitated by the presence of VFF in Portugal. TB’s state of illness, and its long and toxic treatment may demand family and friends’ support. A case study including caregivers of 22 evacuated children (the ones benefiting from bilateral accords) demonstrated how social networks were the main factor relieving them from precarity [66]. Another study using a large individual survey, found that having family and close friends abroad significantly increased the probability of international migration intention [67].

Portugal presents inclusive policies regarding migrants’ health, even if charging variable out-of-pocket fees for service and having eventual access related problems [24,50]. In our study, no participant accessed the CDP directly without a referral, even if direct access is allowed for migrants from a high TB burden country [68]. A Spanish study shows how public healthcare services are less used by irregular migrants than documented migrants or nationals, regardless their country of origin or length of stay [69]. Irregular migrants overusing health services seems a common concern for health systems expressing a universal tendency [70]; however to decrease the burden of infectious disease (classification that follows diagnosis), accessibility should be boosted to reduce diagnostic delays. 

Perceived stigma of TB disease is recognised in many studies [71], and were prominent in participants’ oral statements. A study in Angola showed how emotionally distressed TB patients feel [72]; this fact may dramatically affect health seeking behaviour, diagnostic delays, adherence to treatment, and contact tracing [73,74].Although, a qualitative study among Angolan health professionals, attribute treatment interruptions to patients’ behaviour [57], our interviews show participants feel mistreated by healthcare professionals which may strongly influence linkage to care. 

All countries totalling 20 cases or more (2008–2018), speak Portuguese as their official language, therefore integrating the Community of Portuguese Language Countries–Comunidade dos Países de Língua Portuguesa (CPLP), which aggregate other countries (e.g., Brazil), to the PALOP group. Historical links, diplomatic connections, and a common language, are powerful bonds influencing migrants to historically choose Portugal (or/and Lisbon), as a favourite destination among other countries [75].

We highlight two points raised by this research justifying further research. Firstly, although a high percentage of foreign-born residents in the studied area were from Brazil, its number of notified TB cases was low. Secondly, half of interviewed TB patients did not have a user number. Long TB treatment, and bureaucratic delays may lead migrant TB patients to extend their stay in the host country and become irregular. Most probably these individuals are not counted in the statistics of foreign-born resident citizens. This bureaucratic issue affecting foreign-born patients and its implications, would justify further research.

As a case study, working with absolute numbers was illustrative of TB cases distribution, and its relationship with migration for the selected study site. It perfectly served to underpin a theoretical foray into the topic in a high-risk urban area. The number of interviews may seem limited as many selected TB patients declined the invitation to participate. However, considering that notified TB cases among Angolan and Guinean-Bissauan individuals totalled approximately 40 on a yearly basis, interviews were held with approximately a third of the annual number of patients. Hence, we consider having obtained an adequate number of observations on the subject matter. 

In other big cities in the European context, such as Barcelona, where migration is a recognised risk factor for TB [76], we may encounter similar contexts as the one studied here. However, other large urban areas in Portugal, such as the city of Oporto, present a different disease profile in which other risk factors for TB, such as alcohol consumption, are more significant among notified cases [77].

We set out to deepen the available knowledge on recent migrants’ TB in detriment of autochthonous TB or long-term migrant TB, as it has so far remained under-researched in Portugal. Nevertheless, understanding TB’s pathways for different patient profiles would justify further study. TB in migrants is challenging for health systems because of the complexity inherent to people’s mobility. In depth studies of unequal urban settings and their dynamics, are key to understanding TB disease burdens and to developing locally adapted public health initiatives for better disease control. To our knowledge the distinction between recent and long-term migrants has been rarely explored in the literature. Its exploratory analysis in the present study has shed new light on different TB pathways and patient’s profiles in relation to length of stay in the host country, and countries of origin.

## 5. Conclusions

Tuberculosis is a global health problem. In low-incidence countries it tends to concentrate in specific geographical areas of big cities where an accumulation of risk factors for TB prevails. Notified TB cases in the study setting in the LMA, a critical urban area, are mainly due to autochthonous TB. Nevertheless, foreign-born contribution is relevant, and mainly associated with PALOP countries. By dividing migrants into recent and long-term we aimed to provide a new perspective while hypothesising on TB pathways according to length of stay in the host country. Long-term migrant TB patients’ vulnerability relates to poor social determinants of health associated with the urban environment they live in, which is shared with Portuguese-born TB patients. Recent migrant TB patient’s principal vulnerability is related to the fragile health systems of their origin countries which is a push factor towards migration. Qualitative evidence shows that recent migrant TB patient profile, did not fully correspond to the expected depiction of the vulnerable and impoverished individual. Suffering and diagnostic uncertainty, allied to a close linguistic and family connection with Portugal, a country where healthcare policies are inclusive, were determinant for their pursuit of transnational pathways to seek alternative healthcare and activate new treatment cascades.

## Figures and Tables

**Figure 1 ijerph-19-03834-f001:**
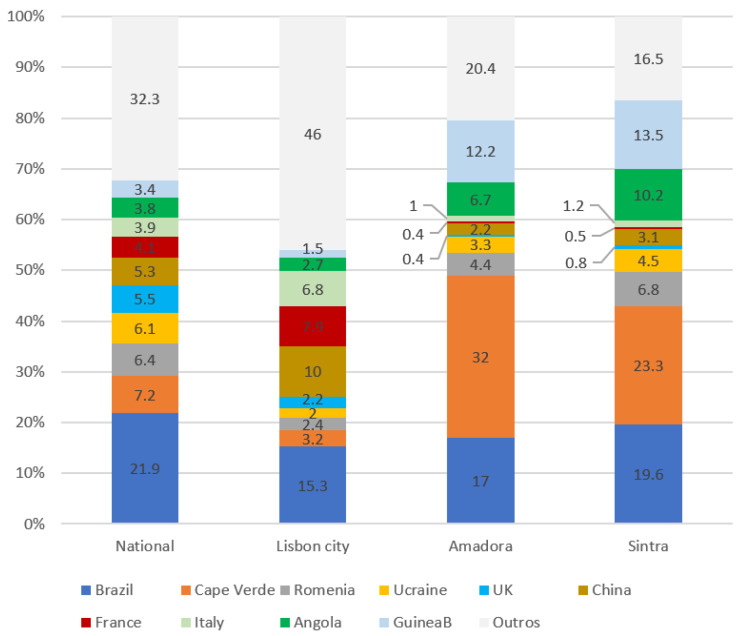
Distribution of nationalities of foreign-born citizens at national level and local level (Lisbon-city, Amadora, and Sintra), in 2018 (Source of data to build graph: SEFSTAT https://sefstat.sef.pt/forms/distritos.aspx, accessed on 10 August 2021).

**Figure 2 ijerph-19-03834-f002:**
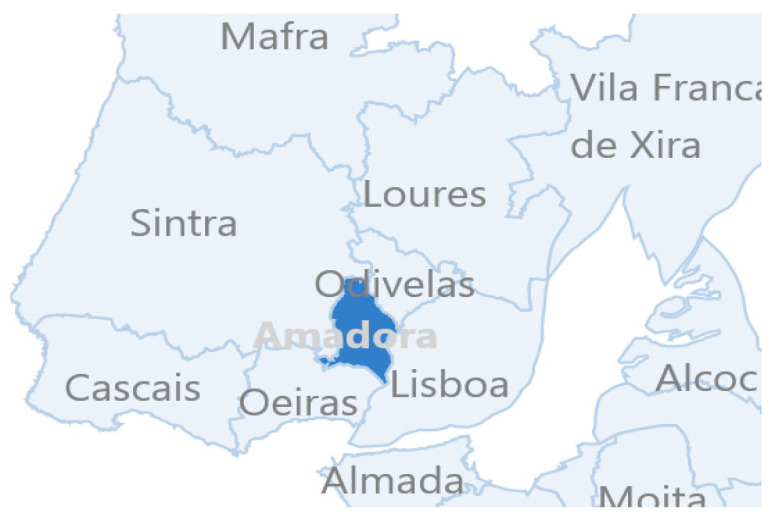
Map of a portion of the LMA area, divided by municipalities. The CDP subject of study serves the area of Amadora, Sintra and a portion of Odivelas. Amadora is highlighted as TB *hotspot* and hosts a *critical urban area*. (Source: https://www.pordata.pt/Municipios accessed 22 October 2021).

**Figure 3 ijerph-19-03834-f003:**
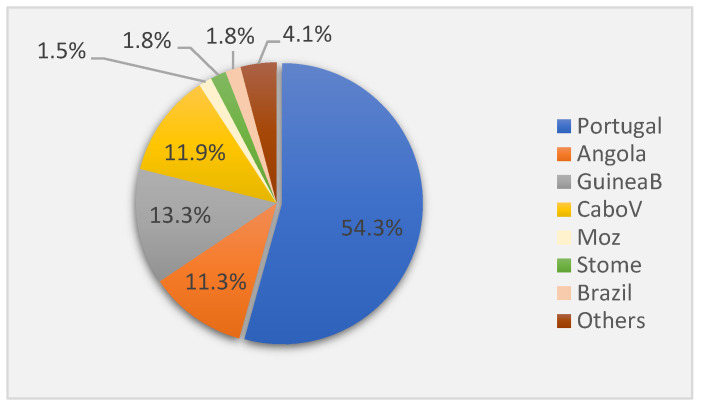
Distribution of notified cases (in %) per country of birth for the study site (2008–2018). GuineaB = Guinea-Bissau; CaboV = Cabo Verde; Moz = Mozambique; STome = São Tomé and Principe.

**Figure 4 ijerph-19-03834-f004:**
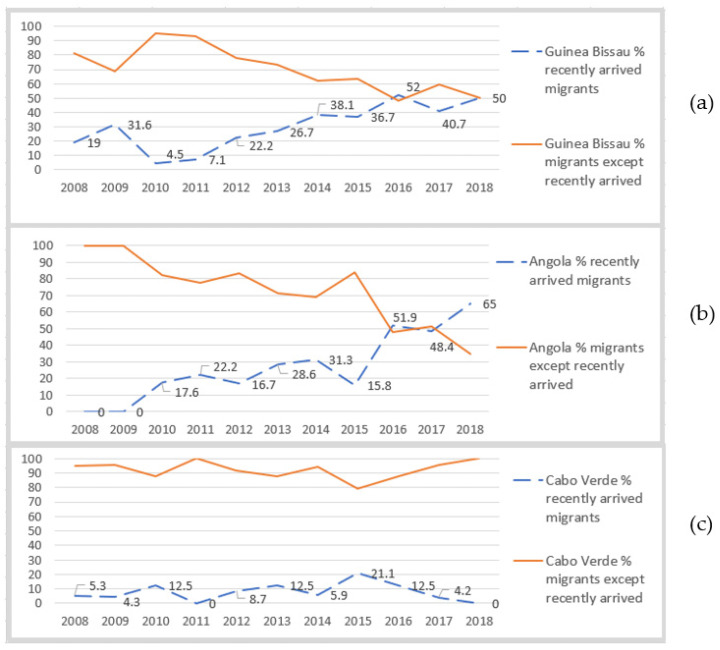
Graphs showing the percentage (%) of notified TB cases in recent migrants (dashed line) mirroring long-term migrants (solid line), for (**a**) Guinea-Bissau, (**b**) Angola, and (**c**) Cabo Verde.

**Table 1 ijerph-19-03834-t001:** TB´s epidemiological indicators for Portugal and the ten countries contributing the most to the number of foreign-born resident citizens in Portugal, in 2018 *.

	Estimated TB Incidence Rate ^+^	# Notified TB Cases	TB-HIV Co-Infection ^+^	TB Treatment Coverage ^	Treatment Success Rate ~
Portugal	16 (13–18)	1445	1.5 (1.2–1.9)	88% (76–100)	71%
Brazil	45 (38–52)	82,930	5 (4.3–5.8)	78% (67–91)	69%
Cabo Verde	39 (30–50)	210	5.1 (3.1–7.6)	95% (75–120)	89%
Romania	64 (54–74)	7698	1.4 (1.1–1.8)	58% (50–69)	84%
Ukraine	73 (49–102)	19,521	16 (11–22)	55% (39–82)	79%
UK	6.9 (6.3–7.6)	4458	0.24 (0.09–0.46)	89% (81–98)	78%
China	59 (50–68)	633,156	0.84 (0.71–0.98)	74% (64–87)	94%
France	8.2 (7.2–9.2)	4606	0.42 (0.32–0.52)	83% (73–94)	12%
Italy	6.6 (5.7–7.6)	2287	0.34 (0.19–0.54)	54% (47–63)	-
Angola	350 (225–503)	66,058	41 (26–59)	55% (38–85)	69%
Guinea-Bissau	361 (234–516)	2561	114 (74–164)	36% (25–55)	89%

* Source: https://www.who.int/teams/global-tuberculosis-programme/data, 2020 data, accessed on 31 January 2022). ^+^ Rate per 100,000 inhabitants; # Absolute Number; ^ Treatment coverage is defined by the number of new and relapse TB cases notified and treated in a given year, divided by the estimated number of incident TB cases in the same year; ~ Treatment success rate is defined as the proportion of cases registered in a given year (excluding cases placed on a second-line drug regimen) that successfully completed treatment without bacteriological evidence of failure.

**Table 2 ijerph-19-03834-t002:** Number of notified TB cases from 2008 to 2018, per country of origin for study site *.

	2008	2009	2010	2011	2012	2013	2014	2015	2016	2017	2018	TOTAL
Portugal	87	94	113	82	100	76	115	112	111	76	65	1031
Guinea-Bissau	21	19	22	28	27	15	21	30	25	27	18	253
Cabo Verde	19	23	24	26	23	24	17	19	16	24	11	226
Angola	13	15	17	18	24	14	16	19	27	31	20	214
S. Tomé	2	4	5	0	4	1	7	3	2	3	3	34
Brazil	2	6	6	3	2	2	4	1	1	6	1	34
Mozambique	2	2	3	4	5	1	4	2	1	1	3	28
Others	3	2	6	7	12	9	11	10	5	5	8	78
TOTAL	149	165	196	168	197	142	195	196	188	173	129	1898

* Countries shown have more than 20 notified cases in total for the considered period.

**Table 3 ijerph-19-03834-t003:** Characteristics of recent migrant interviewees diagnosed with TB.

	Country of Origin	TB Type	Risk Factor	Legal Status	Reason to Come	Entry Point
P1	Angola	P	-	SDD	Health	PHC
P2	Angola	Pleural	-	SDD	Live	Screening
P3	Guinea-B	Ganglionar	HIV	SDD	Health	PHC
P4	Guinea-B	Bone/miliar	HIV	SDD	Health	Private clinic
P5	Angola	Pleural	-	User number	Live/Study	ER
P6	Guinea-B	P	-	User number	Live	ER
P7	Angola	Peritoneal	HIV	User number	Health/Live	PHC
P8	Guinea-B	Disseminated	-	User number	Tourism	Ambulance
P9	Angola	Bone	-	SDD	Health	ER
P10 *	Angola	P	HIV	SDD	Health	PHC
P11	Angola	P	Health p	SDD	Health	Private clinic
P12 *	Angola	Pleural	-	User number	Tourism	PHC
P13	Angola	Pleural	-	User number	Health	Private clinic

* Two participants did not allow audio recording. P = pulmonary TB. Health p = health professional. SDD = irregular situation, only urgent and vital care. PHC = Primary Health Care. ER = Emergency Room.

## Data Availability

Due to ethical restrictions imposed by ethical committees, data are available upon request. To request the data access, readers should contact the Institutional Ethics Committee from the Instituto de Higiene e Medicina Tropical–Universidade Nova de Lisboa, at Rua da Junqueira, 100, 1349-008, Lisbon-Portugal (comissaodeetica@ihmt.unl.pt) to request permission, and copying in the following authors into the request mirandaribeiro.rafaela@gmail.com and isabelc@ihmt.unl.pt. However, the part of the data used in the manuscript underlying the findings is freely available in the tables and appendix.

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
