# Peer review of "Tuberculosis and Migrant Pathways in an Urban Setting: A Mixed-Method Case Study on a Treatment Centre in the Lisbon Metropolitan Area, Portugal"

_ijerph, 2022, doi:10.3390/ijerph19073834_

Round 1
Reviewer 1 Report
This is an article describing a mix-method study to investigate TB in urban settings in Lisbon, Portugal.
Major comment:
1) Reduce the introduction section as it now includes 140 lines and more than 40 references;
2) Some definitions used in the introduction are vague (i.e., "recently arrived migrants", or simply "migrants"), although concepts in the section are far to be summarized. This affects the conclusion when comparing policies of the proportion of a subset in different populations (e.g. lines 188-194);
3) Reduce the length of the discussion. Having so many words for a few key concepts isn't helpful for the Reader;
4) Authors reiterate the concept that TB is due to many socioeconomic factors in their study although they do not provide any evidence;
5) One of the main results of the study is the variation shown in Figure 4. However, the authors only touch this point in their manuscript and they do not comment on it enough.
Minor comments:
- Line 174 define PHC
- Figure 2: percentages in the figure cannot be read in some cases
- The English require an in-depth revision of the manuscript
- Remove any redundancy in the contents of the manuscript. Try to be concise;
- Results (lines 290-299): how were groups defined? Is this grouping approach needed?
- Lines 307-308: a temporal association does not mean causation: if someone from a country X arrives in Portugal in 2008 and then move each year, let's say in the summer, back to the country of origin and in 2012 develops TB this might be "imported" and not "acquired" in Portugal.
- Lines 529-534: consider also undetected TB cases in migrants due to barriers in accessing health care.
Author Response
"Please see the attachment."

Reviewer 2 Report
This manuscript describes a qualitative and quantitative combined study assessing the impact of migrant status and its drivers on the incident cases of TB in a high-risk part of Urban Portugal. It is a very well-conceived study and addresses important gaps in the body of knowledge on the subject. The Methods and Results section are very well written and warrant very few comments except for technical rationale of the quantitative section which I cannot comment about.
Minor comments
The introduction can be improved considerably as per comments below, but by and large these are minor.
- LINE 39;” …. associated with a greater likelihood of developing TB in infected individuals” please correct this sentence.
- Line 48-9: standard treatment for EPTB is still 6 months except in certain organs with poor drug penetration. The authors should revise this statement to avoid confusion and implication that EPTB is treated longer in general.
- “Thirty countries concentrate around 90% of the TB disease burden - none of them a high-30 income country” This sentence seems to be out of place in page 1st paragraph and seems to be more appropriate in page 2, 2nd
- Line 58-9 “a migration gradient is tendentially south-north...” Please check the accuracy and context of this statement. Globally Asia and Russia have 1st and 3rd largest numbers of international migrants in 2019-2020 according to IOM and these are not in the Southern Hemisphere.
- “Portugal has presented an increasing number of TB cases of foreign origin, from 18.3% to 63 23.7%, from 2016 to 2019; while total TB cases decreased for the same period, from 1836 to 64 1771, respectively” Please simplify this sentence for clarity. Are the authors talking about absolute numbers and percentages for Portugal for those years?
- Line 69-70 “Infectious diseases, such as TB or HIV, have the potential to affect host countries´ epidemiology, representing a worrisome public health issue” The last part of this sentence can be more precise.
- “TB can be brought into host countries, by migrants who developed TB [19], however, it may also result from an activation of a previously acquired latent infection in the context of immunosuppression [20] or be the result of a new infection in the host country [21].” The authors can improve and use this sentence to support the point made above in comment 6. The last parts of the sentence seem to be out of context.
- “The resident migrant population in Portugal increased between 2015 and 2019, with a surplus of 34.1% during a four-year period, totaling 590.348 persons [26]. Can the authors simplify this sentence? Are they suggesting that between 2015-9 resident migrant population increased from XXXX to XXXX, an increase of 590.348 persons (34.1%)?
- Line 104 check grammar
- Line 108: The largest urban areas of Portugal - Lisbon and Oporto - accounted for 57,3% of the national TB burden in 2017 [30]. It would be useful to compare this with the proportion of the total country population the two metros contribute
- “Lisbon presented the largest number of multi-drug resistant TB cases in Portugal.” Same comment as above,
- For instance, in Oporto unemployment was a relevant risk factor, while in Lisbon the immigrant population from high burden country was more relevant [34]. Please add numbers for comparison and strengthen the argument
- Line 127 “(infection or disease)” Please clarify the difference between these two. Are the authors referring to latent TB? Please clarify what is a “delimited area”
- Line 148 This despite the fact that Portugal presents inclusive policies when compared to other EU countries, affording irregular migrants “more than minimum rights”[44]. This statement uses a relatively old reference and adds little to the message the authors are driving across and I suggest removing it. There are newer studies for example that compare focus on inclusivity vs quality in EU and focus varies between the two in various countries.
- It would be useful to refer to Fig 2 earlier as mentioned above in the comment as it answers some of the comments raised above.
- Line 215: “Among possible risk factors associated with the study area, this paper focused on migrants, for two reasons.” This is not clear
- Expand OECD
Major comments
- Line 85: “The most representative country was Brazil, accounting for 21.9% of migrant residents in 2018, followed by Cabo Verde, Romania and Ukraine with 7.2%, 6.4% and 6.1% respectively.” The word “representative” is misplaced. Secondly are these numbers referring to the total number of resident migrants or just for the period 2015-9? If the former maybe the authors should swap this with the preceding statement which is narrower.
- Line 100-6 Similar comments as above to contextualize the statements. . It would be useful to tabulate this data rather than picking only a few countries to highlight coverage disparities in the migrant country of origin.
- Line 90-9. This paragraph’s context can be improved by having stated in the introduction which countries contribute the 90% either as a table or mentioning the most relevant ones to the discussion as well as their contribution. It would be useful to tabulate. (This is already in Figure 2 which can be referenced here.
- One of the novel findings which the authors seem to underemphasize in their conclusions is the direct role the healthcare system in the country-of-origin vs the number of incident cases in that country. It would also add value to briefly compare or hypothesize on why Mozambique and Angola differ so much e.g., Angola with a lower HIV-TB co-infection rate vs Mozambique with a much healthier TB treatment coverage (notified/estimated incidence) and the relative influence of these.
- The Discussion can be organized better to highlight the major findings of the study first and the other issues later. The major findings are hidden amongst a lot of other information that is less impactful. It may be a good idea to list these findings in the first paragraph of the discussion and serially expand each of them followed by the less important findings.
Author Response
"Please see the attachment."

Round 2
Reviewer 1 Report
The manuscript has greatly improved from its previous version.
I have still a few comments easy to address:
- The authors have used in-depth interviews by selecting representatives from the "recent migrant" category. In the discussion (line 547), however, they generalize that the healthcare systems in the countries of origin (and transit) were weak. In the manuscript, there are also other types of "migrants" (or, better, foreign-born) not coming from the African continent. I would also soften your thesis by using "was/were considered weak...".
- The above applies to all conclusions drawn from the semi-structured interviews (e.g. migratory routes, etc.).
Author Response
"Please see the attachment."
